# How do care home staff understand, manage and respond to agitation in people with dementia? A qualitative study

Penny Rapaport,[1,2] Gill Livingston,[1] Olivia Hamilton,[1] Rebecca Turner,[1] Aisling Stringer,[1] Sarah Robertson,[1,2] Claudia Cooper[1]

[1]UCL Department of Old Age Psychiatry, Division of Psychiatry, University College London, London, UK
[2]North Thames CLAHRC, London, UK

**Correspondence to**
Dr Penny Rapaport;
p.rapaport@ucl.ac.uk

## ABSTRACT

**Objectives** Little is known about how care home staff understand and respond to distress in residents living with dementia labelled as agitation. The aim of this study was to describe how care home staff understand and respond to agitation and the factors that determine how it is managed.

**Design** We conducted a qualitative thematic analysis.

**Setting** We recruited staff from six care homes in South East England including residential and nursing homes of differing sizes run by both the private and charity sector and located in urban and rural areas.

**Participants** We interviewed 25 care home staff using purposive sampling to include staff of either sex, differing age, ethnicity, nationality and with different roles and experience.

**Results** We identified four overarching themes: (1) behaviours expressing unmet need; (2) staff emotional responses to agitation; (3) understanding the individual helps and (4) constraints on staff responses. Staff struggled with the paradox of trying to connect with the personhood of residents while seeing the person as separate to and, therefore, not responsible for their behaviours. Staff often felt powerless, frightened and overwhelmed, and their responses were constrained by care home structures, processes and a culture of fear and scrutiny.

**Conclusions** Responding to agitation expressed by residents was not a linear process and staff faced tensions and dilemmas in deciding how to respond, especially when initial strategies were unsuccessful or when attempts to respond to residents' needs were inhibited by structural and procedural constraints in the care home. Future trials of psychosocial interventions should support staff to identify and respond to residents' unmet needs and include how staff can look after themselves.

## INTRODUCTION

Over 70% of UK care home residents have dementia,[1] often with complex needs and high levels of behavioural and psychological symptoms of dementia (BPSD).[2] The term BPSD describes a heterogeneous range of symptoms including apathy, irritability, anxiety, depression, psychosis and agitation. As a syndrome, BPSD has been criticised as poorly defined, with authors calling for a focus on specific symptoms and targeted interventions.[3] In this study, we have, therefore, focused on the most common of these symptoms, agitation.[4] While agitation is variously defined, the term is often used to refer to a range of behaviours, including restlessness, pacing, repetitive vocalisations and verbally or physically aggressive behaviour.[2 5] Agitation in care homes is associated with lower quality of life and higher care costs[6–8] and is persistent and distressing.[8 9]

In a recent epidemiological care home study, we found that although severity of agitation was associated with severity of dementia, this was not a linear association, with 45% of those with moderate and severe dementia experiencing clinically significant levels of agitation. From this, we concluded that the agitation cannot be fully explained in terms of worsening brain pathology,[8] reiterating the need to also conceptualise agitation in social

**Strengths and limitations of this study**

► To the best of our knowledge, this is the first study to qualitatively explore how staff in care homes understand and respond to behaviours labelled as agitation in people living with dementia.
► Understanding how staff understand and manage agitation and what makes this harder or easier can inform the development of a psychosocial intervention that reflects and addresses the complexity of delivering interventions in care home settings.
► We recruited staff based on manager's recommendations or existing research team and staff relationships which may have resulted in recruitment bias.
► We conducted semistructured interviews rather than directly observing care interactions; therefore, our analysis reflects our interpretation of staff perspectives.

and psychological terms. Kitwood highlights the relational nature of personhood and outlines how negative interactions between carers and people with dementia create a 'malign social psychology' undermining personhood and resulting in unmet social and psychological needs. Conversely, interventions that address these needs and promote personhood may prevent and reduce manifestations of distress such as agitation.[10]

Being labelled as agitated may increase the difficulties individuals with dementia face by impacting on personhood; how they are perceived, understood and responded to.[11 12] This can have real consequences for people living with dementia in care homes, for example, by resulting in increased use of restraint[13 14] and increased prescribing of psychotropic medications.[15] Antipsychotic medication has limited effect in reducing symptoms of agitation and leads to increased morbidity and mortality in people with dementia.[16] Other medications have limited efficacy and significant harmful side effects.[16–19] Evidence for non-pharmacological alternatives to manage agitation is mixed, with few interventions demonstrating effects after the intervention is completed.[20 21]

The need-driven, dementia-compromised behaviour (NDB) theory[22] proposes that behaviours in dementia, often labelled as disruptive, arise from unmet needs. Needs may be emotional (communication, comfort or contact), recreational (stimulation and enjoyable activities) and physical (pain relief, thirst, hunger or treatment of constipation or infection). Environmental limitations can prevent needs being met, when staff are unavailable, unaware or inadequately skilled in communicating and interacting with people with dementia. Care home staff often have little training and are low paid, with high staff turnover.[23 24] Communication between staff and residents, for example, during personal care, can be dominated by instructions.[25]

Further understanding of the relational aspects of agitation in care homes, of how staff make sense of and respond to agitation, is necessary to facilitate development of more effective and sustainable interventions. To our knowledge, this is the first qualitative study of care home staff experiences of caring for residents with dementia experiencing agitation. We aim to describe how care home staff understand and respond to agitation and the factors that determine how it is managed.

## METHODS
### Setting, participants and procedures
We purposively selected care homes participating in Managing Agitation and Raising QUality of LifE in Dementia (MARQUE), a study involving people with dementia living in care homes[8] interviewing staff from varied care home settings: residential and nursing, differing sizes, private and charity sector, and in urban and rural areas in Southeast England. All of the homes we approached agreed to participate. We included staff providing direct care and support to residents with

dementia, including care assistants, senior carers (who had additional responsibilities), team leaders, activities coordinators, registered nursing staff and managers. We did not interview staff in solely domestic, catering or administrative roles. We sampled purposively to ensure that we interviewed staff of either sex, differing age, ethnicity, nationality and with different roles and experience. We used a semistructured interview schedule (see online supplementary appendix 1) based on the literature, consultation with dementia family carers and research team expert opinion. Recruitment and data collection procedures are outlined in figure 1.

### Patient and public involvement
People whose lives have been affected by dementia were members of the project management group and were involved in the development of the research questions for the project. To inform the development of the interview schedule, we held a focus group for family carers who had cared for a relative with agitation in a care home, discussing from their perspective, what we needed to consider in our interviews with care home staff. After completing the analysis of the data, we conducted follow-up focus groups presenting the findings and discussing how this may inform intervention development.

### Data analysis
We took an inductive thematic analytical approach based on the work of Braun and Clarke.[26] After completing each set of interviews (in one care home), PR listened to the recordings, reflected on initial themes and revised the interview schedule to incorporate new ideas expressed by care staff, and as part of an ongoing reflective process based on both the emerging perspectives of the participants and the interviewer.[27] This also allowed us to check that the questions made sense to the participants, especially since over half did not speak English as a first language.

PR and one of four independent raters (OH, RT, AS and SR) systematically coded each transcript into meaningful fragments and labelled these initial codes, discrepancies were discussed and resolved.[28] PR, GL and CC then organised the data into preliminary themes, making connections between codes, then displaying in matrices and diagrams developing a comprehensive picture of the phenomena in question. We discussed the coding frames within the team using the constant comparison method,[29] identifying similarities and differences in the data and refining our account in an iterative process closely grounded in the data. We ceased interviews at thematic saturation, at the point that neither of the two researchers coding an interview identified new codes and when the authors' reflections on additional interviews resulted in no further emergent themes. PR, CC, GL and OH agreed this by consensus. We sought respondent validation by sending participants summaries of the findings, allowing them to comment on the accuracy and credibility of interpretations[30] (see online supplementary appendix 2). The

Research assistants with existing relationships with the care homes approached the care home managers and explained the purpose of the interviews, asking if they were happy for PR to contact them to discuss this further.

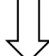

PR contacted managers to discuss the interviews and to arrange to interview staff during their shifts without impacting on care provision or staff breaks; the study budget covered replacement staff costs so that staff could participate in interviews.

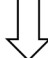

With manager's agreement, the researchers approached individual staff members, explaining the purpose of the study and providing information sheets. If they wished to participate interviews were arranged at a convenient time.

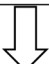

PR conducted interviews in private rooms in the care homes and as part of obtaining informed written consent, went through the information sheets again reiterating that participation was voluntary.

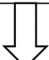

PR explained to participants the limits of confidentiality in accordance with the NRES approval and reiterated that they could stop the interview at any point or take a break, using clinical skills to set up a comfortable and safe space for discussion and to put participants at ease.

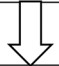

All interviews were digitally recorded and the recorder returned to the university after interviews. The audio files were uploaded and stored on a secure network after which the audio files were deleted from the recorders.

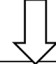

Audio files were securely transferred and transcribed verbatim by a professional transcription company. Identifying information was removed to preserve anonymity and transcripts were password protected; on completion of the analysis all recordings were deleted.

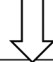

Identifying information was removed and transcripts password protected; on completion of the analysis all recordings were deleted. PR listened to each interview to check the transcription and entered all transcribed interviews into NVivo 11 software for qualitative analysis.

**Figure 1** Outline of recruitment and data collection procedures. NRES, National Research Ethics Service; PR, Penny Rapaport.

Standards for Reporting Qualitative Research [31] checklist is presented in online supplementary appendix 3.

## RESULTS
### Study participants
PR interviewed staff in six care homes in London (n=4), Kent (n=1) and Cambridge (n=1) between July 2014 and January 2015. Five homes were private and one was run by a charity. Three were nursing homes, two residential homes and one provided residential and nursing care. Table 1 summarises sociodemographic and employment characteristics of staff interviewed.

### Qualitative findings
We identified four overarching themes: (1) behaviours expressing unmet need; (2) staff emotional responses to agitation; (3) understanding the individual helps and (4)

**Table 1** Care staff sociodemographic and employment characteristics

| Sociodemographic | Category | Care staff n (%) |
|---|---|---|
| Gender | Female | 17 (68) |
| | Male | 8 (32) |
| Ethnicity | Asian or Asian British | 6 (24) |
| | Black or black British | 6 (24) |
| | White British | 6 (24) |
| | White other | 5 (20) |
| | Mixed other | 2 (8) |
| English as first language | No | 13 (52) |
| | Yes | 11 (44) |
| | Not known | 1 (4) |
| Staff role | Care assistant | 9 (36) |
| | Senior carer | 5 (20) |
| | Manager/deputy manager | 5 (20) |
| | Team leader/unit manager | 2 (8) |
| | Activities coordinator | 2 (8) |
| | Registered general nurse | 2 (8) |
| Shift pattern | Days | 18 (72) |
| | Days and nights | 7 (28) |
| Length of service | Less than 1 year | 4 (16) |
| | 1–5 years | 13 (52) |
| | 6–10 years | 8 (32) |
| Nursing qualification | Yes | 10 (40) |
| | No | 15 (60) |

constraints on staff responses. These are discussed below and presented in online supplementary appendix 4.

### Behaviours expressing unmet need

Staff identified a range of behaviours as expressions of unmet emotional, physical or environmental needs. Staff experienced behaviours as persistent, unpredictable and inter-related. The behaviours most commonly described were verbal and physical aggression expressed when staff provided intimate care, such as washing, dressing or assisting a resident to the toilet.

> Like when you do the personal care he just goes with you for a while and then suddenly he'll react, it's like if you are not taking care of yourself the carers can be hit or sometimes those kinds of things, he'll just smash you like that. (Team leader; Home 1)

### Unmet physical need

Many staff described how residents were often unaware or unable to communicate that they were in pain or unwell.

> Maybe they have pain in their legs, but they can't explain themselves. That is why I say you need patience,

because they're shouting, they don't know where the pain is. (Senior carer; Home 4)

Identifying the cause of distress involved a process of elimination and consideration of various factors such as hunger, thirst or medication. Senior staff commented that these causes may be overlooked by care staff.

> Someone might be shouting out and instead of asking, are you in pain, or investigating why, you'll find a lot of, well, I'll do you a cup of tea. (Deputy manager; Home 2)

### Unmet emotional need

Staff drew on knowledge of residents' past and present to understand how behaviours might arise from emotional distress. Staff conceptualised unmet emotional needs in terms of residents feeling frustrated, feeling ashamed or embarrassed during personal care and feeling insecure or abandoned. Some staff felt that physical needs were viewed as more valid or more deserving of staff responses than emotional needs, especially when working under pressure.

> Unfortunately, what it comes to is you start to think to yourself, well, these people have real needs…because she wants the companionship, but, you know, we do have a hierarchy in terms of -is the person pain free, are they well hydrated, are they fed well, are they comfortable, and then maybe you can get to the social needs. (Deputy manager; Home 2)

Some staff perceived overt expressions of distress such as repeated screaming or calling out as attention seeking or intentionally demanding.

> It's a bit of a game sometimes for him I think. There's a lot of play-acting getting involved. This guy probably has lot more capacity than he thinks. (Care assistant; Home 3)

### Unmet environmental need

Staff described how residents may be distressed both due to a lack of interaction and under or overstimulation.

> Just the whole idea of them sitting in a chair is no good. I don't like that. The brain must be stimulated, even if it is a small amount. (Care assistant; Home 3)

> Like this one, who doesn't like any noise, he will stand up straightaway, like when he is having lunch, he will bang the cutlery on the table and then look at the person who is screaming. (Care assistant 2; Home 5)

Staff also believed that care home environments contributed to unmet needs.

> If you walk into the lounge in a care home it isn't like your home. There isn't a sofa, there's single chairs, and who has single chairs? And something that small can make a big part on someone, especially if

someone is affectionate; they want to sit next to somebody. (Care assistant; Home 2)

A number of staff described residents feeling trapped, evoking images of imprisonment.

He finds this home that it's a prison. (Activities coordinator; Home 3)

I can open the door. I can have a walk outside. It's not for them. They are always going, either in this left corridor, or to the far end of the right one, or in the lounge, or in the dining area. That's it. Finished. (Care assistant; Home 4)

### Staff emotional responses to agitation
Staff sometimes struggled to respond to residents' behaviours, especially if more than one resident was involved. They reported being unable to give space to a resident or let them express discomfort, as they wanted to minimise the impact on others.

So it can be very difficult if he shouts all night. It's not fair on them because he's disrupting somebody and they don't sleep. Alright fair enough, he's got his got his own problems but what about the other residents. (Senior carer; Home 4)

### Feeling powerless and disheartened
Staff frequently described feeling powerless, especially when attempts to alleviate a resident's distress were not working, or if they could not make sense of a resident's behaviour. This was particularly so when residents displayed persistent, repetitive behaviours.

But to have someone distressed in front of you, then…someone else is getting distressed… They are looking at you to try and do something, and you can't do anything. (Deputy manager; Home 5)

At these times, staff judged themselves as not 'doing a good job' and feeling judged by others undermining their professional identity.

It can make you feel sometimes, when things aren't working, that you've failed…sometimes you do go home disarmed, because you feel that you haven't been able to do the best for that person. (Activities coordinator; Home 2)

### Feeling frightened
Staff also expressed fear of being harmed. This connected to feelings of powerlessness, especially when residents hit or shouted at staff. Anticipating harm affected how staff approached and responded to residents.

They are scared. It doesn't mean they don't do it, but, you know what I mean? While you're doing things, you're not doing with all the openness and things; you do it with an 'oohf'. (Care assistant; Home 4)

Staff narrated these behaviours as 'part of the job', yet highlighted how difficult it was fearing for their own safety.

Sometimes it is quite traumatic to be slapped or to be kicked or to be scratched or…you know, it's not an easy thing to say, okay, I'll brush it off. (Deputy manager; Home 5)

### Trying not to react
Although staff described resident behaviours as unintentional, they sometimes reacted in ways they regretted. They described trying not to react to aggression, the effort required to stay calm and how their immediate reactions could escalate behaviours.

It may make you react in a way that you don't want to, because you know these residents can't help their behaviours, but…you're stressed…and you may say something…you shouldn't say, or…raise your voice at a resident, which you…shouldn't do, but at that moment, you're thinking, oh, no, again. (Nurse; Home 2)

I think being calm is a big thing, and not reacting because, when you're getting smacked in the face, you know, some people's natural reaction would be to say something. (Care assistant; Home 2)

### Understanding the individual helps
Staff found that having time to get to know and understand residents was critical to building trust and familiarity with residents. This helped them to understand and respond to residents' needs.

### Seeing the person not the disease
Staff described what they termed a person-centred approach as getting to know the person with dementia informing their responses to distress behaviours. They talked about seeing residents as equals and imagining how they would feel in their situation. Staff drew on notions of shared humanity to describe how they maintained empathy and compassion, connecting with the personhood of individuals.

I think they should be able to come in, yes, do the personal care, but while you're doing the personal care, look at the rest of the person, not only the bit you're washing and dressing, remember that they're a human being. (Unit manager; Home 3)

However, staff also described behaviours that may be construed as socially unacceptable, like swearing or displaying aggression, as part of the dementia and separate to the person, moving between these different, arguably contradictory positions. This tension was apparent when staff tried to talk directly to residents about behaviours considered unacceptable. Seeing people with dementia as like themselves led some to feel that they might be able to control their behaviour and were, therefore, to blame for it.

We said that you have to apologise to your wife because it was not nice…swearing at her. So…maybe he realise but he say, I don't want to, I'm not going to apologise…Maybe he just doesn't remember…when he was swearing. (Care assistant; Home 5)

### Connecting with previously valued identities

Knowing about a person's past and using this during care was viewed as a respectful way to calm residents.

I always like to know what did you used to do in your time. What work do you like doing, you know. All the different things really, in life. (Activities coordinator; Home 3)

Sometimes relatives facilitated this process by sharing information and explaining 'what works'. Staff often perceived behaviours as expressing distress at losing independence and spoke of supporting individuals to reconnect with what was important to them.

[It] is a way of showing your independence… So giving her the money that she can pay with, she feels that she's paid…and that…she's worthy to have that sort of thing. (Activities coordinator; Home 2)

### Playing along rather than correcting

Staff struggled to know how to respond when residents were disoriented, especially when residents were unaware of the extent of their dependency. Staff at all levels talked about 'playing along' or entering their reality as better than trying to orient people.

They may not be in the here and now, but let's go back to where they are, it's very interesting when you go back to where they are…if they feel they're a teenage girl, well, okay, we talk like a teenage girl. (Care assistant; Home 2)

This was not simple and did not always have a positive effect. Staff found it hard to decide when to stop 'playing along'. They felt uncomfortable lying to residents as it could increase confusion.

While you standing arguing with someone saying, no, you're 90 and your kids are all grown up. To them they're…still at school…You wouldn't go as far as saying, oh, they've just gone to the shop. They'll be back in a minute because then that minute they could still be, well, where is she? (Deputy manager; Home 6)

### Making people feel comfortable and at home

Making people feel at home involved creating a stimulating and comforting environment. Staff described trying to comfort residents, particularly more impaired residents, using music, touch and other sensory stimulation, in addition to activities led by specialist staff.

It's a 24 hours process and this is their home, they can get up when they like, as long as they eat and they feel comfortable, that's the most important. (Senior carer; Home 5)

They talked about how touch made a big difference to residents, otherwise only touched during personal care.

I'll say to him, do you want to dance? Because he liked to dance. He'll take me really close and we'll have a little bit of a dance. (Care assistant; Home 2)

### Constraints on staff responses
### Procedural constraints

As noted above, getting to know the residents and delivering person-centred care and was a preferred approach, however, many acknowledged that care delivery was divided into a series of tasks, with an inherent tension between task-focused and person-focused approaches.

Changing in that [person-centred] direction is very difficult, because people start thinking, oh, if I do that, I might get told off. If I do that, then I won't be able to fill in the dishwasher by quarter past 11, or if I do this instead of that, then they're going to tell me off because I didn't take the bin, so it's all this kind of balancing act. (Deputy manager; Home 5)

Additionally, there seemed to be an implicit hierarchy of how staff should respond to residents' needs, prioritising basic needs over a need for company or interaction. Staff related this to feeling that they did not have the time to engage residents in activities, relying on activities coordinators for this.

And you might be doing an activity with someone, the guy in one of the rooms pressed the emergency… he is almost like needs one to one care so you might be rushed off to attend to him really. It really is, the activities really does demand an extra carer I think. (Care assistant; Home 3)

### Structural constraints

Staff in all homes commented on financial challenges facing the sector, describing a business culture incompatible with delivering personalised care, particularly when it reduced staffing levels and therefore time. Staff also spoke about how, in a home that was part of a larger company, they felt anonymous and disconnected from the wider organisation. One staff member recalled having her glasses broken by a resident and the company refusing to pay for repair. Other staff said that minimum wage pay, antisocial shift patterns and staffing levels make it harder to maintain compassion.

Sometimes it can be challenging because if the budget doesn't meet…then the staff need to be reduced, and… The needs of the residents take second place. (Deputy manager; Home 5)

And, I think, with these big homes where there are 109, 110 beds, it's too much… It is just a conveyor belt; who's next, who's next, who's next? (Deputy

manager; Home 6)

## Support and training

Staff described feeling devalued by managers and not heard or taken seriously when they raised concerns. Where they felt unappreciated by residents and relatives, appreciation by managers took on additional importance.

> You don't always feel valued for the job that you're doing; it is a very difficult job. It does have an effect… on your working practise…caring for people all day and it doesn't always feel as though staff are really cared for. (Unit manager; Home 3)

In most homes staff described how they would 'keep it to ourselves', seeking support from their immediate team, assuming that managers would be unhelpful. This response was perhaps heightened by feeling that managers cannot understand their experiences as frontline carers and will not provide solutions.

> Well, even if I told somebody, I don't know what they could do. What could they do? (Care assistant 2; Home 3)

Although many care staff spoke of a lack of managerial support, they (and the managers) also highlighted examples of good practice. Hands-on managers and feeling that managers had done or the job themselves left staff feeling that managers could understand their difficulties. Staff also highlighted how learning from peers and seniors through discussion and joint working enabled them to find new ways to respond.

> I have had training [but] I've gone to management and they've taught me a different way to try and cope with it, I feel being there, dealing with it, doing it, is the best training. (Care assistant; Home 2)

## Culture of fear and scrutiny

Staff felt that the media focused on negative aspects of care, particularly abuse and neglect. They thought media overlooked the good practice that they saw, as well as the impact, particularly of behaviours that in other contexts would be construed as abusive, from residents towards staff, eroding staff morale.

> Sometimes it would be so lovely to hear a nice story about dementia, and staff, and what people do, and… You don't hear things about residents lashing out at carers. (Care assistant; Home 2)

In some cases, negative perceptions of care homes cultivated a fear of making mistakes or getting into trouble. This stifled more creative and flexible approaches to meeting residents' unmet needs. Staff felt that appearances were sometimes prioritised over minimising distress, for example, insisting a resident changed a dirty top or came out of their room.

> There's the cover your back kind of fear to people… I think that translates back into the negative thing where you don't want to try a new thing in case it hurts someone or it puts them at risk (Deputy manager; Home 2)

> And they'll say, why is my mum being in bed? And, you know…obviously we tried our best and…it does annoy. We're always writing it down and just inform the Nurse so we don't get in trouble (Care assistant 2; Home 4)

## DISCUSSION

### Main findings

To our knowledge, this is the first qualitative study to explore how care home staff experience, understand and respond to behaviours labelled as agitation in people with dementia and what helps or hinders their responses. The findings indicate that staff in care homes understand behaviours labelled as agitation as multifaceted and relational, consistent with conclusions of the MARQUE cross-sectional study that agitation is not entirely explained in terms of brain pathology.[8]

The findings support the NDB theory with staff explaining agitation as expressions of unmet needs in residents.[22] Even if staff engage in a process of trial and error and do not fully understand what is causing a particular behaviour, this process of sense making encourages them to take a curious and person-centred position in relation to those they are caring for. It may also highlight the range of behavioural responses available to them, rather than leaving staff feeling that nothing can be done; reinforcing that finding ways to address these needs by getting to know the individual can prevent or reduce agitation.[10]

Understanding the needs of residents was not straightforward for staff and although some staff felt that unmet physical needs could be viewed as more valid than emotional and social needs, they were also felt to be frequently overlooked. Existing research has found that pain and discomfort is underdetected in those with severe dementia in care homes and that discomfort is associated with higher levels of verbal aggression.[32] This is important since in the presence of behaviours perceived as aggressive, staff felt powerless and frightened, impacting on their capacity to respond to resident underlying needs. These findings are consistent with an existing study that found behaviours perceived as aggressive, uncooperative and unpredictable were felt to be most difficult to manage.[33]

Staff wanted to deliver person-centred care, but struggled to do this when feeling overwhelmed, unsupported by management and unsafe or fearful. They faced tensions in deciding how far to go along with a resident's disorientation or how to separate a person from their behaviour without undermining personhood. In his work on personhood and dementia, Kitwood highlighted the relational dimension of personhood as connected to both 'cared for' and 'carer'.[12] Generally, however, this has been

related to how those caring for people with dementia can enhance or diminish personhood through their responses and ultimately this may result in staff being blamed or seen as the cause of problems by not being person centred or doing a good enough job.

The impact of structural and procedural factors on staff well-being and care practices has previously been documented qualitatively[34–37] and quantitatively.[24 38] Consistent with this, staff here indicated that they internalised a culture of scrutiny and fear from within and outside of care homes. This sometimes prevented staff from trying new approaches and staff felt that the care home sector was increasingly incompatible with an individualised approach. This is concerning given that inappropriate treatment of people with dementia in residential care often occurs when staff feel unable to meet clients' needs,[39] possibly because it results in emotional distancing in the context of more institutionalised care. This fits with our recently published cross-sectional survey on abuse in care homes, where abusive/neglectful behaviour was more common in homes where staff experienced more burn-out and feelings of depersonalisation towards people with dementia.[40]

### Clinical implications

These findings have implications for the development of sustainable and practical interventions which build on approaches that staff find useful and address the practical and structural constraints discussed above. These findings reinforce the need to find ways to support staff to manage their own emotional responses and reactions when residents are agitated, as well as supporting them to reflect systematically on recognising and meeting residents' unmet needs.

### Limitations

Although our sampling meant that we accessed a breadth of viewpoints, contributing to the richness of this account, we directly approached staff based on manager's recommendations or existing research team and staff relationships. There may, therefore, be an inherent recruitment bias. Staff could have felt pressured into taking part, which is why PR spent time before taking consent reiterating the voluntary nature of participation, answering any questions about the process, giving staff the opportunity to change their mind. In relying on interviews with care home staff, we present their narration of their experiences and perceptions of how they deliver care. We must be cautious not to overgeneralise our findings.

### Future research

This study highlights the complex inter-relationship between notions of personhood and needs-driven behaviours labelled as agitation[11] building on research highlighting high skill levels required by staff expected to deliver person-centred care in care homes, and the complexity of achieving this.[35] Future research should explore the long-term impact of interventions designed to reduce agitation, which incorporate this complexity, on care home culture and if and how they become embedded in care practices.

**Acknowledgements** We wish to thank all the care home staff who participated in this study and all the Alzheimer's Society research network volunteers, whose lives have been affected by dementia, for their contribution. Thank you to Joanna Murray for her contribution to the qualitative analysis.

**Contributors** All authors made a substantial contribution to this work. PR, CC and GL contributed to the conception and design of the study and PR drafted the paper. All authors critically revised it and gave final approval for this version to be published. PR collected all the data and coded all the interview transcripts. OH, RT, AS and SR coded some of the interview transcripts. PR, CC and GL then organised the data into preliminary themes.

**Funding** This study was funded by a grant from the UK Economic and Social Research Council and the National Institute of Health Research Grant number NIHR/ESRC ES/L001780/1. CC and GL are supported by the UCLH NIHR Biomedical Research Centre. This research was supported by the National Institute for Health Research (NIHR) Collaboration for Leadership in Applied Health Research and Care North Thames at Bart's Health NHS Trust (NIHR CLAHRC North Thames).

**Disclaimer** The views expressed are those of the author(s) and not necessarily those of the NHS, the NIHR or the Department of Health. The funders had no role in study design, data collection and analysis, decision to publish or preparation of the manuscript.

**Competing interests** None declared.

**Patient consent** Not required.

**Ethics approval** London (Queen's Square) National Research Ethics Service (NRES) committee gave approval (reference: 14/LO/0697) for the study.

**Provenance and peer review** Not commissioned; externally peer reviewed.

**Data sharing statement** The data will be made available by the authors on request.

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
