## [Reviewer comments · BMJ Open]

ARTICLE DETAILS

TITLE (PROVISIONAL)	How do care home staff understand, manage and respond to agitation in people with dementia? A qualitative study
AUTHORS	Rapaport, Penny Livingston, Gill Hamilton, Olivia Turner, Rebecca Stringer, Aisling Robertson, Sarah Cooper, Claudia

VERSION 1 - REVIEW

REVIEWER	Richard Gray
REVIEW RETURNED	13-Mar-2018

GENERAL COMMENTS	Thank you for inviting me to review this paper. The authors report a qualitative study of care staff experience managing resident agitation. This is an interesting topic and I thought the reporting was good. There were some specific areas where I felt that the authors could provide more detail and/or add clarity. Specifically: - Do the authors need to make a distinction between "agitation" and BPSD. Perhaps make a distinction between the two. I am worried that the concepts were blurred when participants were talking about their experiences- Could the authors give a definition of "care home staff"?- I did not think that the study objective was clearly expressed. The sentence "what they find works or obstructs good practice" was quite "clunky".- Abstract. Could the study design simply be described as "qualitative thematic analysis".- Care homes were in the South of England (rather than England)?- Provide details of number of care homes asked to participate, number who responded to the invitation, etc.- When reporting themes/sub-themes can you number 1, 1a, 2 etc. It would make it easier to follow.- Was there a single overarching meta theme?- Abstract. The conclusion did not flow logically from the results.- Page 3. Please set out the risks of antipsychotic use in this population. You assume this knowledge it would be informative if the state of knowledge was summarised.- Page 3. What are the alternatives to antipsychotics?- Page 3. Avoid the use of the word "posits"- You state "Interventions promoting better communication..." Can you provide details of the Livingstone et al
---

	trial and Jutkowitz et al review? As a minimum study design and sample size.  - You tend to use quotation marks to "emphasize a point". Can I discourage you from doing this? - "This is the first..." You need to be more circumspect. Perhaps, to the best of our knowledge this is... - I did not think the rationale for the study was as robust as it could have been. The authors may wish to consider revising. - More precise detail about how participants were recruited is required. - Details of the consent procedure need to be given - The authors should consider in more detail the ethical considerations of the study. For example, could participants have felt pressured into taking part. How was confidentiality assured (including in reporting). - More detail of the process of approaching participants needs to be given. - Give detail of how audio recordings were stored and transferred (from care home to University, was the data recorder ever taken home for example). - Can some justification be given to the authors decision to listen to interviews and revise interview schedules? - I'm not convinced that data saturation had been achieved. Provide more detail of how you made this decision. I wondered if this should be in the results section. - Data analysis. I think the authors need to explain with more precision/clarity how data were analyzed. For example, when you say "using the constant comparison method" what does this mean. There should be sufficient detail that would enable me to replicate - Will data be available for reanalysis? If not why not? - Table 1. Two participants were nurses. Do you mean registered nurses (1st or 2nd level)? - I found the tables in the results section disrupted the flow of the narrative. - Can you report the number of the theme/sub-theme in the sub-heading? - CH, presumably care home? Perhaps avoid abbreviations. - I'm not keen on single quotes to exemplify a sub-theme, worth reviewing? - Page 7, line 50. Don't put xxx it scans as though it was an error. - Were interviews conducted in work time?
--	--

REVIEWER	Professor Christine Øye
REVIEW RETURNED	22-Mar-2018

GENERAL COMMENTS	Review: How do care home staff understand, manage and respond to agitation in people with dementia? A qualitative study Introduction: The introduction focuses very much on agitation which is okey since this is the article's core theme. However, agitation can be understood in different ways based on bio-medical, social or
--

	psychological explanations. This can be further outlined, since there is a mix of such references. It is also a bit unclear how unmet needs are defined since this is a core finding in the study. Needs are very well defined, but unmet needs are less outlined and defined. Unmet needs are very well outlined and defined by Kitwood (1997). Also, references to person-centred approaches in relation to unmet needs can be helpful for the discussions section in relation to core findings page 9 (Understanding the individual helps). Moreover, a lot of the references is based on a more bio-medical understanding of dementia, while the empirical examples and the aims for the article is more in line with a person-centred understanding of dementia. Therefore, I think there is a mismatch between some of the introduction literature and the empirical findings presented. I do not understand why the authors link how front-line staff understand and respond to residents experiencing agitation, and how this kind of knowledge can develop more effective interventions. Rather, the author argue for not “standardized interventions” at the end of the article, but use of real-life examples facilitating change. There might be a confusion in relation to interventions and how qualitative findings can be used in such intervention approaches. This can be clarified, or the authors can consider to delete a discussion in relation to interventions since this is not stated as a the aim of the article. I also think that a discussion on intervention or change approaches would need a whole new article. Furthermore, I think that this study is relevant in relation to avoid use of restraint which is very common in relation to agitation. References to publications which raise the links between agitation and use of restraint is missing, and which can show the high relevance of this study. Method No comments Results Under the section: Understanding the individual helps, I miss some empirical examples of how a person-centred approach can calm down an agitated person, since the core theme in the article is about agitation. Discussion Overall, the discussion sums up the main findings of the study and relates the findings to some of the background literature. Furthermore, the article relates some of the findings with structural and procedural factors, also documented in this study. However, I think that the article can discuss the findings further for instance by “using” literature on unmet needs and person-centred care since these themes are core findings in the study. The aim of the study is descriptive while the discussion is in the last part explanatory. I would think that the article could add an aim of how the findings should be discussed, in relation to what kind of theory (person-centred care etc.) or as it is now in relation to structural factors
--	---

VERSION 1 – AUTHOR RESPONSE

Reviewer 1: Richard Gray

Thank you for inviting me to review this paper. The authors report a qualitative study of care staff experience managing resident agitation. This is an interesting topic and I thought the reporting was good. There were some specific areas where I felt that the authors could provide more detail and/or add clarity. Thank you, we address these specific areas below.

Do the authors need to make a distinction between "agitation" and BPSD. Perhaps make a distinction between the two. I am worried that the concepts were blurred when participants were talking about their experiences.

We have added the following text to the introduction:

"The term BPSD describes a heterogeneous range of symptoms including apathy, irritability, anxiety, depression, psychosis and agitation. As a syndrome, BPSD has been criticised as poorly defined, with authors calling for a focus on specific symptoms and targeted interventions.¹ In this study, we have therefore focused on the most common of these symptoms, agitation.²"

Although we defined agitation in our study information and interview schedule, we agree that there was blurring in the participants discussion of their experiences, highlighting the complexity and diversity of both the experiences and language used by staff. We have amended the discussion of clinical implications to address this, adding to the text:

"Although the term agitation is widely used in clinical and academic settings, it represents a "thin description"³ whereas staff described a diverse range of behaviours that were broader than the definition of agitation. This reflects the rich complexity of how they experience and interpret these behaviours in care homes. Therefore, the intervention conceptualises behaviours as expressions of unmet needs in residents and introduces heuristics and practical activities, focused upon staff developing in-depth understanding of resident distress."

Could the authors give a definition of "care home staff"?

Yes, we have added to the methods section:

"We included staff providing direct care and support to residents with dementia, including care assistants, senior carers (who had additional responsibilities), team leaders, activities coordinators, registered nursing staff and managers. We did not interview staff in solely domestic, catering or administrative roles."

I did not think that the study objective was clearly expressed. The sentence "what they find works or obstructs good practice" was quite "clunky" and I did not think the rationale for the study was as robust as it could have been. The authors may wish to consider revising.

We have clarified the stated objective in the abstract to:

"The aim of this study was to describe how care home staff understand and respond to agitation and the factors that determine how it is managed."

We have changed the introduction to clarify the rationale and aim of the study. We have added at the end of the introduction:

"Further understanding of the relational aspects of agitation in care homes; of how staff make sense of and respond to agitation, is necessary to facilitate development of more effective and sustainable interventions. To our knowledge, this is the first qualitative study of care home staff experiences of caring for residents with dementia experiencing agitation. We aim to describe how care home staff understand and respond to agitation and the factors that determine how it is managed. We plan to

use the findings to inform the development of a sustainable and effective intervention to support staff in care homes to manage agitation.”

Abstract. Could the study design simply be described as "qualitative thematic analysis".

Yes, we have changed the abstract to say: “We conducted a qualitative thematic analysis.”

Care homes were in the South of England (rather than England)?

This is true and we have amended the abstract and methods to say the care homes were in the South East of England.

Provide details of number of care homes asked to participate, number who responded to the invitation, etc.

Because we recruited care homes that were already participating in the MARQUE (Managing Agitation and Raising QUality of LifE in Dementia) study and with whom our team had pre-existing relationships all of the homes we approached agreed to take part in the qualitative study. We have added this to the text in the methods section:

“All of the homes we approached agreed to participate.”

Was there a single overarching meta theme?

No, there was not a single overarching meta-theme.

Abstract. The conclusion did not flow logically from the results.

We have amended the abstract and think that the logical flow is now improved. We added the following text:

“Responding to agitation expressed by residents was not a linear process and staff faced tensions and dilemmas in deciding how to respond, especially when initial strategies were unsuccessful.”

Page 3. Please set out the risks of antipsychotic use in this population. You assume this knowledge it would be informative if the state of knowledge was summarised.

Apologies, we have added the following text to the introduction:

“Antipsychotic medication has limited effect in reducing symptoms of agitation and leads to increased morbidity and mortality in people with dementia.⁴ Other medications are also problematic, with limited efficacy and significant harmful side effects.⁴⁻⁷”

Page 3. What are the alternatives to antipsychotics? And you state "Interventions promoting better communication..." Can you provide details of the Livingstone et al trial and Jutkowitz et al review? As a minimum study design and sample size.

We have added in further information on the evidence for non-pharmacological alternatives to anti-psychotics and further detail on the Livingstone et al paper. Due to the limited word count we have not included sample sizes of the individual studies included in the systematic review however this information and detail on study quality and the specific interventions is included in the referenced review. The text we have added/amended is:

“Evidence for non-pharmacological alternatives in managing agitation is mixed, with few interventions demonstrating effects after the intervention is completed.^{8 9} In a systematic review of RCTs testing non-pharmacological interventions for agitation, Livingstone et al (2014) found that group based activities, music therapy by protocol and sensory interventions, decrease agitation whilst the

intervention is being delivered, and supervised interventions training staff in communication skills and person-centred care, reduce agitation for up to six months post-intervention.”

Page 3. Avoid the use of the word "posits"

We have changed the word “posits” to “proposes”.

You tend to use quotation marks to "emphasize a point". Can I discourage you from doing this?

Yes, of course, we have changed this throughout the manuscript.

"This is the first..." You need to be more circumspect. Perhaps, to the best of our knowledge this is...

We have changed the text to: “To our knowledge this...”

More precise detail about how participants were recruited is required and more detail of the process of approaching participants needs to be given.

We have added more detail on recruitment procedures in figure 1, which has been added to the manuscript. The following text has been included in the figure:

“Research assistants with existing relationships with the care homes approached the care home managers and explained the purpose of the interviews, asking if they were happy for PR to contact them to discuss this further.”

“PR contacted managers to discuss the interviews and to arrange to interview staff during their shifts without impacting on care provision or staff breaks; the study budget covered replacement staff costs so that staff could participate in interviews.”

“With manager’s agreement, the researchers approached individual staff members, explaining the purpose of the study and providing information sheets. If they wished to participate interviews were arranged at a convenient time.”

Details of the consent procedure need to be given and the authors should consider in more detail the ethical considerations of the study. For example, could participants have felt pressured into taking part. How was confidentiality assured (including in reporting).

We have added the following text into figure 1:

“PR conducted interviews in private rooms in the care homes and as part of obtaining informed written consent, went through the information sheets again reiterating that participation was voluntary.”

“PR explained to participants the limits of confidentiality in accordance with the NRES approval and reiterated that they could stop the interview at any point or take a break, using clinical skills to set up a comfortable and safe space for discussion and to put participants at ease.”

We have also added in and amended the limitations section of the discussion to say:

“Staff could have felt pressured into taking part, which is why PR spent time before taking consent reiterating the voluntary nature of participation, answering any questions about the process, giving staff the opportunity to change their mind.”

Give detail of how audio recordings were stored and transferred (from care home to University, was the data recorder ever taken home for example).

We have added to the text into figure 1:

“All interviews were digitally recorded and the recorder returned to the university after interviews. The audio files were uploaded and stored on a secure network after which the audio files were deleted from the recorders.”

“Audio files were securely transferred and transcribed verbatim by a professional transcription company. Identifying information was removed to preserve anonymity and transcripts were password protected; on completion of the analysis all recordings were deleted.”

Can some justification be given to the authors decision to listen to interviews and revise interview schedules?

Yes, this was part of an ongoing reflective process whereby we revisited our research questions and processes throughout the qualitative enquiry based both upon the participants’ responses to the interview and the interviewers own experiences and understandings of the process. We have amended the following text and moved it to the data analysis section of the methods:

“After completing each set of interviews (in one care home), PR listened to the recordings, reflected on initial themes and revised the interview schedule to incorporate new ideas expressed by care staff, and as part of an ongoing reflective process based upon both the emerging perspectives of the participants and the interviewer.¹⁰ This also allowed us to check that the questions made sense to the participants, especially since over half did not speak English as a first language.”

I'm not convinced that data saturation had been achieved. Provide more detail of how you made this decision. I wondered if this should be in the results section.

We reached thematic saturation, which we defined as no new codes emerging from additional interviews according to two researchers and as a team we did not feel that further themes were emerging in relation to our research questions. We have kept this in the methods rather than the results section for the flow of the account. We have amended the text to provide more detail on the process:

“We ceased interviews at thematic saturation, at the point that neither of the two researchers coding an interview identified new codes and when the authors’ reflections on additional interviews resulted in no further emergent themes. PR, CC and GL and OH agreed this by consensus.”

Data analysis. I think the authors need to explain with more precision/clarity how data were analyzed. For example, when you say "using the constant comparison method" what does this mean. There should be sufficient detail that would enable me to replicate

We have added more specific detail and references for the data analysis and clarified what we mean by the constant comparison method:

“We took an inductive thematic analytic approach based upon the work of Braun and Clarke. ¹¹ ...”PR, GL and CC then organised the data into preliminary themes, making connections between codes, then displaying in matrices and diagrams developing a comprehensive picture of the phenomena in question. We discussed the coding frames within the team using the constant comparison method, ¹² identifying similarities and differences in the data and refining our account in an iterative process closely grounded in the data.”

Will data be available for reanalysis? If not why not?

Yes the data will be available for reanalysis on request, I have added a data sharing statement to the manuscript.

Were interviews conducted in work time?

Yes, we have added this information to the methods section.

Table 1. Two participants were nurses. Do you mean registered nurses (1st or 2nd level)?

Yes they were employed as registered general nurses in nursing home settings. We have added this to the text in table 1. We do not have further information on whether they were 1st or 2nd level nurses.

I found the tables in the results section disrupted the flow of the narrative. We have moved the list of themes (Table 2) to an appendix and removed Table 3, integrating additional quotes from this table into the text. We have left the table of demographics (Table 1) as we feel this is useful information and does not disrupt the narrative flow of the qualitative results.

Can you report the number of the theme/sub-theme in the sub-heading?

Yes, we have added this to the manuscript.

CH, presumably care home? Perhaps avoid abbreviations.

We have changed this in the text.

I'm not keen on single quotes to exemplify a sub-theme, worth reviewing?

We have amended this and each sub-theme now has more than one quote.

Page 7, line 50. Don't put xxx it scans as though it was an error.

Apologies, we have changed this.

Reviewer 2: Professor Christine Øye

Introduction:

The introduction focuses very much on agitation which is okey since this is the article`s core theme. However, agitation can be understood in different ways based on bio-medical, social or psychological explanations. This can be further outlined, since there is a mix of such references. It is also a bit unclear how unmet needs are defined since this is a core finding in the study. Needs are very well defined, but unmet needs are less outlined and defined. Unmet needs are very well outlined and defined by Kitwood (1997). Also, references to person-centred approaches in relation to unmet needs can be helpful for the discussions section in relation to core findings page 9 (Understanding the individual helps). Moreover, a lot of the references is based on a more bio-medical understanding of dementia, while the empirical examples and the aims for the article is more in line with a person-centred understanding of dementia. Therefore, I think there is a mismatch between some of the introduction literature and the empirical findings presented.

Thank you for your comments, we agree that agitation can be understood in biological, psychological and social terms and that this will have consequences for how care staff respond and intervene. We have amended the introduction and added references to make this distinction more explicit. We have added the text:

“In a recent epidemiological care home study, we found that although severity of agitation was associated with severity of dementia this was not a linear association, with 45% of those with moderate and severe dementia experiencing clinically significant levels of agitation. From this, we concluded that the agitation cannot be fully explained in terms of worsening brain pathology,13 reiterating the need to also conceptualise agitation in social and psychological terms. Kitwood (1997) highlights the relational nature of personhood and outlines how negative interactions between carers and people with dementia create a ‘malign social psychology’ undermining personhood and resulting

in unmet social and psychological needs. Conversely, interventions that address these needs and promote personhood may prevent and reduce manifestations of distress such as agitation.¹⁴”

I do not understand why the authors link how front-line staff understand and respond to residents experiencing agitation, and how this kind of knowledge can develop more effective interventions. Rather, the author argue for not “standardized interventions” at the end of the article, but use of real-life examples facilitating change. There might be a confusion in relation to interventions and how qualitative findings can be used in such intervention approaches. This can be clarified, or the authors can consider to delete a discussion in relation to interventions since this is not stated as a the aim of the article.

In line with the MRC framework for the development and testing of complex interventions we believe that in the early stages of intervention development, in-depth qualitative work exploring the target of interventions can usefully inform both the content and processes of intervention delivery.¹⁵ In relation to agitation, there is limited evidence for sustained effects of non-pharmacological interventions aimed at changing care home practices. Therefore, we feel that there is a need to understand both the context in which care staff are responding to agitation, and their experiences of doing so, in order to develop an intervention that explicitly addresses these complexities. We have amended the introduction to clearly outline this rationale, reordering and adding to the original text. Specifically we have added:

“Further understanding of the relational aspects of agitation in care homes; of how staff make sense of and respond to agitation, is necessary to facilitate development of more effective and sustainable interventions. To our knowledge, this is the first qualitative study of care home staff experiences of caring for residents with dementia experiencing agitation. We aim to describe how care home staff understand and respond to agitation and the factors that determine how it is managed. We plan to use the findings to inform the development of a sustainable and effective intervention to support staff in care homes to manage agitation.”

I also think that a discussion on intervention or change approaches would need a whole new article.

We agree that a further article on change approaches is warranted and will do so in relation to our process evaluation of the pilot intervention and main trial.

Furthermore, I think that this study is relevant in relation to avoid use of restraint which is very common in relation to agitation. References to publications which raise the links between agitation and use of restraint is missing, and which can show the high relevance of this study.

Thank you, we have added in the following text to the introduction:

“Being labelled as agitated may increase the difficulties individuals with dementia face by impacting upon personhood; how they are perceived, understood and managed by others.^{16 17} This can have real consequences for people living with dementia in care homes, for example by resulting in increased use of restraint^{18 19} and increased prescribing of psychotropic medications.²⁰”

Method - No comments

Results

Under the section: Understanding the individual helps, I miss some empirical examples of how a person-centred approach can calm down an agitated person, since the core theme in the article is about agitation.

We have added in additional quotes for each of the subthemes providing further empirical examples of how a person-centred approach can calm down an agitated person. These additions are:

Seeing the person not the disease:

I think they should be able to come in, yes, do the personal care, but while you're doing the personal care, look at the rest of the person, not only the bit you're washing and dressing, remember that they're a human being. (Unit manager; Home 3)

Connecting with previously valued identities:

I always like to know what did you used to do in your time. What work do you like doing, you know. All the different things really, in life... (Activities coordinator; Home 3)

Playing along with rather than correcting:

They may not be in the here and now, but let's go back to where they are... if they feel they're a teenage girl, well, okay, we talk like a teenage girl. (Care assistant; Home 2)

Making people feel comfortable and at home:

It's a 24 hour process and this is their home, they can get up when they like, as long as they eat and they feel comfortable, that's the most important. (Senior carer; Home 5)

Discussion

Overall, the discussion sums up the main findings of the study and relates the findings to some of the background literature. Furthermore, the article relates some of the findings with structural and procedural factors, also documented in this study. Thank you.

However, I think that the article can discuss the findings further for instance by "using" literature on unmet needs and person-centred care since these themes are core findings in the study.

We have added additional text to the discussion to address this omission:

"The findings are consistent with the NDB theory. 21 A potential benefit of this approach, is that even if staff engage in a process of trial and error and do not fully understand what is causing a particular behaviour, this process of sense making encourages them to take a curious and person-centred position in relation to those they are caring for. Perhaps what is absent from the staff experiences is the opportunity to reflect systematically upon what is happening, as they are so immersed in it. This may explain why approaches to managing agitation that focus on communication skills 22, structured reflection 23 and delivering individualised person-centred care 24 have so far been most effective." And "This is concerning given that inappropriate treatment of people with dementia in residential and day care often occurs when staff feel unable to meet clients' needs,25 possibly because it results in emotional distancing in the context of more institutionalised care. This fits with our recently published cross-sectional survey on abuse in care homes, where abusive/neglectful behaviour was more common in homes where staff experienced more burnout and feelings of depersonalisation towards people with dementia. 26"

The aim of the study is descriptive while the discussion is in the last part explanatory. I would think that the article could add an aim of how the findings should be discussed, in relation to what kind of theory (person-centred care etc.) or as it is now in relation to structural factors

We have clarified the aim / objectives of the study and hopefully that it is now clear that there is a descriptive and an explanatory dimension. We have also added the following text to the discussion to align with the revised manuscript.

"Our aims were to describe staff experiences and to offer an explanatory framework to inform the development of a non-pharmacological intervention aimed at supporting staff in care homes to manage agitation."

Formatting amendments: Please re-upload your supplementary files in PDF format.

We have now done this.

Having made the above revisions our paper's word count is 4798, exceeding the recommended word count recommended in your 'instructions to authors'. In order to ensure the qualitative results and discussion are clear, we have not been able to condense the article to the desired word limit. We look forward to hearing from you.

Dr Penny Rapaport (on behalf of all the authors)

References

1. Selbaek G. Behavioural and psychological symptoms in dementia. *Tidsskrift for den Norske laegeforening: tidsskrift for praktisk medicin, ny raekke* 2005;125(11):1500-02.
2. Margallo-Lana M, Swann A, O'Brien J, et al. Prevalence and pharmacological management of behavioural and psychological symptoms amongst dementia sufferers living in care environments. *Int J Geriatr Psychiatry* 2001;16(1):39-44.
3. Geertz C. *Thick Description: Towards an Interpretative Theory of Culture. The Interpretation of Cultures: Selected Essays.* New York: Basic Books 1973:3-30.
4. Fox C, Crugel M, Maidment I, et al. Efficacy of memantine for agitation in Alzheimer's dementia: a randomised double-blind placebo controlled trial. *PLoS One* 2012;7(5):e35185. doi: 10.1371/journal.pone.0035185
5. Howard RJ, Juszcak E, Ballard C, et al. Donepezil for the treatment of agitation in Alzheimer's disease. *N Engl J Med* 2007;357(14):1382-92. doi: 10.1056/NEJMoa066583
6. Husebo BS, Ballard C, Sandvik R, et al. Efficacy of treating pain to reduce behavioural disturbances in residents of nursing homes with dementia: cluster randomised clinical trial. *BMJ* 2011;343:d4065. doi: 10.1136/bmj.d4065
7. Porsteinsson AP, Drye LT, Pollock BG, et al. Effect of citalopram on agitation in Alzheimer disease: the CitAD randomized clinical trial. *JAMA* 2014;311(7):682-91. doi: 10.1001/jama.2014.93
8. Livingston G, Kelly L, Lewis-Holmes E, et al. Non-pharmacological interventions for agitation in dementia: systematic review of randomised controlled trials. *Br J Psychiatry* 2014;205(6):436-42. doi: 10.1192/bjp.bp.113.141119
9. Jutkowitz E, Brasure M, Fuchs E, et al. Care-Delivery Interventions to Manage Agitation and Aggression in Dementia Nursing Home and Assisted Living Residents: A Systematic Review and Meta-analysis. *J Am Geriatr Soc* 2016;64(3):477-88. doi: 10.1111/jgs.13936
10. Agee J. Developing qualitative research questions: a reflective process. *International Journal of Qualitative Studies in Education* 2009;22(4):431-47. doi: 10.1080/09518390902736512
11. Braun V, Clarke V. Using thematic analysis in psychology. *Qualitative Research in Psychology* 2006;3(2):77-101. doi: 10.1191/1478088706qp063oa
12. Glaser BG. The Constant Comparative Method of Qualitative-Analysis. *Social Problems* 1965;12(4):436-45. doi: DOI 10.1525/sp.1965.12.4.03a00070
13. Livingston G, Barber J, Marston L, et al. Prevalence of and associations with agitation in residents with dementia living in care homes: MARQUE cross-sectional study. *BJPsych Open* 2017;3(4):171-78. doi: 10.1192/bjpo.bp.117.005181
14. Kitwood TM. *Dementia reconsidered: The person comes first.* Open university press 1997.
15. Moore GF, Audrey S, Barker M, et al. Process evaluation of complex interventions: Medical Research Council guidance. *BMJ* 2015;350:h1258. doi: 10.1136/bmj.h1258
16. Higgs P, Gilleard C. Interrogating personhood and dementia. *Ageing Ment Health* 2016;20(8):773-80. doi: 10.1080/13607863.2015.1118012
17. Kitwood T, Bredin K. Towards a theory of dementia care: personhood and well-being. *Ageing Soc* 1992;12(03):269-87. doi: 10.1017/S0144686X0000502X

18. Hantikainen V. Nursing staff perceptions of the behaviour of older nursing home residents and decision making on restraint use: a qualitative and interpretative study. *Journal of Clinical Nursing* 2001;10(2):246-56.
19. Evans D, FitzGerald M. Reasons for physically restraining patients and residents: a systematic review and content analysis. *International Journal of Nursing Studies* 2002;39(7):735-43.
20. Department of Health. The use of antipsychotic medication for people with dementia: Time for action. A report for the Minister of State for Care Services by Professor Sube Banerjee. , 2009.
21. Algase DL, Beck C, Kolanowski A, et al. Need-driven dementia-compromised behavior: An alternative view of disruptive behavior. *American Journal of Alzheimer's Disease* 2016;11(6):10-19. doi: 10.1177/153331759601100603
22. McCallion P, Toseland RW, Lacey D, et al. Educating nursing assistants to communicate more effectively with nursing home residents with dementia. *Gerontologist* 1999;39(5):546-58. doi: 10.1093/geront/39.5.546
23. Lichtwarck B, Selbaek G, Kirkevold O, et al. Targeted Interdisciplinary Model for Evaluation and Treatment of Neuropsychiatric Symptoms: A Cluster Randomized Controlled Trial. *The American journal of geriatric psychiatry : official journal of the American Association for Geriatric Psychiatry* 2018;26(1):25-38. doi: 10.1016/j.jagp.2017.05.015
24. Chenoweth L, King MT, Jeon YH, et al. Caring for Aged Dementia Care Resident Study (CADRES) of person-centred care, dementia-care mapping, and usual care in dementia: a cluster-randomised trial. *The Lancet Neurology* 2009;8(4):317-25. doi: 10.1016/S1474-4422(09)70045-6
25. Sormunen S, Topo P, Eloniemi-Sulkava U, et al. Inappropriate treatment of people with dementia in residential and day care. *Aging Ment Health* 2007;11(3):246-55. doi: 10.1080/13607860600963539
26. Cooper C, Marston L, Barber J, et al. Do care homes deliver person-centred care? A cross-sectional survey of staff-reported abusive and positive behaviours towards residents from the MARQUE (Managing Agitation and Raising Quality of Life) English national care home survey. *PLOS ONE* 2018;13(3):e0193399. doi: 10.1371/journal.pone.0193399

VERSION 2 – REVIEW

REVIEWER	Richard Gray
REVIEW RETURNED	14-May-2018

GENERAL COMMENTS	The authors have addressed all the points that I raised in my previous review
---

REVIEWER	Christine Øye
REVIEW RETURNED	07-May-2018

GENERAL COMMENTS	Thank you for letting me review this study, which offers an insight into the troublesome domain of working with persons living with dementia in institutional age care settings, especially when residents are agitated. The article is part of a larger project aiming to understand and help staff struggling with challenging behavior among residents living with dementia. The descriptive aim of this article is easy to follow and understand, and the results of the article reflect very well the main aim of the article. Nevertheless, the article has another practical aim to base an intervention on the descriptive qualitative findings. The description of this planned intervention is unclear described and hard to follow also in the discussion section, since this is a planned activity and not an activity which reflects the findings – for instance we do not get any
---

	results from the staffs` experiences with the intervention. Neither any findings or discussion on facilitating or hindering factors explaining the interventions success or failure. Therefore, I think that this article should concentrate itself on the results on the articles main aim, and further outline explanations in regard to the main findings in the discussion section. For instance, how can the authors explain and understand main findings in relation to unmet needs? Further, the discussion section is a bit incomplete, and focuses a lot on how to overcome or cope with agitation based on different initiatives or interventions, and less on how to relate the discussion on the findings. Perhaps this has to do with the mixing up of this descriptive study with a planned intervention study further down the line. Nevertheless, clinical and practical implications is important but does not need to fill the main space for the discussion section – also since the article has a section for this. If the authors concentrate on explaining the main findings in the discussion section and not on the planned intervention, the background literature references from intervention studies should be removed. However, a lot of relevant studies are cited in the background section (and only a few studies are referred to in this section) as well as theory on person-centred care (Kitwood) which can be further used in the discussion section to deepen the discussion. Also, the abstract and summary must be slightly changed. In the discussion section a new aim of the article was added (not outlined in the introduction section), to offer an explanatory model – despite that the article did not offer a theory or model in the method section (probably because the study is based on an inductive model). However, a descriptive inductive qualitative study can use theory as well as background literature to further explain or understand it`s findings, see comments above. Minor comments It is not stated how the researchers collected informed consent, how anonymity was secured or other research ethical guidelines were followed despite a stated ethical approval.
--	---

VERSION 2 – AUTHOR RESPONSE

Reviewer: 2

Reviewer Name: Christine Øye

Institution and Country: Western Norway University of Applied Sciences, Norway Competing

Interests: none declared

Thank you for letting me review this study, which offers an insight into the troublesome domain of working with persons living with dementia in institutional age care settings, especially when residents are agitated. The article is part of a larger project aiming to understand and help staff struggling with challenging behavior among residents living with dementia. The descriptive aim of this article is easy to follow and understand, and the results of the article reflect very well the main aim of the article.

Thank you.

Nevertheless, the article has another practical aim to base an intervention on the descriptive qualitative findings. The description of this planned intervention is unclear described and hard to follow also in the discussion section, since this is a planned activity and not an activity which reflects the findings – for instance we do not get any results from the staffs` experiences with the intervention.

Neither any findings or discussion on facilitating or hindering factors explaining the interventions success or failure.

As you note above, this preliminary qualitative study was part of a larger programme of work that aims to reduce agitation and improve quality of life in people with dementia. This qualitative study was in part intended to inform the future development of more sustainable and usable psychosocial interventions (including the intervention we were planning to develop as part of the MARQUE programme). As such, in line with the MRC framework for the development and testing of complex interventions¹, we have used this qualitative analysis to develop a greater understanding of how staff are currently managing agitation and what makes this process harder or easier.

We apologise if the description of the planned intervention in the discussion section is unclear or we overly focused on the intervention development and we have amended both the introduction and discussion accordingly. We have removed from the introduction the second part of the aim stating:

“We plan to use the findings to inform the development of a sustainable and effective intervention to support staff in care homes to manage agitation.”

And from the discussion:

“Our aims were to describe staff experiences and to offer an explanatory framework to inform the development of a non-pharmacological intervention aimed at supporting staff in care homes to manage agitation.”

We have also deleted from the discussion the existing discussion of clinical implications:

“We have built our findings into the MARQUE intervention, and are currently testing it in a randomised controlled trial. Although the term agitation is widely used in clinical and academic settings, it represents a “thin description”² whereas staff described a diverse range of behaviours that were broader than the definition of agitation. This reflects the rich complexity of how they experience and interpret these behaviours in care homes. Therefore, the intervention conceptualises behaviours as expressions of unmet needs in residents and introduces heuristics and practical activities, focused upon staff developing in-depth understanding of resident distress. Many strategies highlighted as useful in this study have been incorporated, such as: using tools to get to know residents better; offering comfort during routine care; allowing time for staff to share ideas and find ways to manage their own feelings. External barriers have been addressed by delivering the intervention to all staff, involving managers, developing action plans, encouraging staff to use and reflect upon new approaches, offering ongoing supervision, and leaving materials for new staff.”

And replaced this section with a briefer discussion of clinical implications:

“Clinical implications

These findings have implications for the development of sustainable and practical interventions which build upon approaches that staff find useful and address the practical and structural constraints discussed above. These findings reinforce the need to find ways to support staff to manage their own emotional responses and reactions when residents are agitated, as well as supporting them to reflect systematically on recognising and meeting residents’ unmet needs.”

Therefore, I think that this article should concentrate itself on the results on the articles main aim, and further outline explanations in regard to the main findings in the discussion section. For instance, how can the authors explain and understand main findings in relation to unmet needs?

Further, the discussion section is a bit incomplete, and focuses a lot on how to overcome or cope with agitation based on different initiatives or interventions, and less on how to relate the discussion on the findings.

We have restructured and added to the discussion on the main findings, particularly in relation to staff understandings of unmet need. We have also reduced the focus on different interventions and initiatives to cope with agitation. We have added in:

“The findings indicate that staff in care homes understand behaviours labelled as agitation as multi-faceted and relational, consistent with conclusions of the MARQUE cross sectional study that agitation is not entirely explained in terms of brain pathology.³

The findings support the NDB theory with staff explaining agitation as expressions of unmet needs in residents. ⁴ Even if staff engage in a process of trial and error and do not fully understand what is causing a particular behaviour, this process of sense making encourages them to take a curious and person-centred position in relation to those they are caring for. It may also highlight the range of behavioural responses available to them, rather than leaving staff feeling that nothing can be done; reinforcing that finding ways to address these needs by getting to know the individual can prevent or reduce agitation.⁵

Understanding the needs of residents was not straightforward for staff and although some staff felt that unmet physical needs could be viewed as more valid than emotional and social needs, they were also felt to be frequently overlooked. Existing research has found that pain and discomfort is under-detected in those with severe dementia in care homes and that discomfort is associated with higher levels of verbal aggression.⁶ This is important since in the presence of behaviours perceived as aggressive, staff felt powerless and frightened, impacting upon their capacity to respond to resident underlying needs. These findings are consistent with an existing study that found behaviours perceived as aggressive, uncooperative and unpredictable were felt to be most difficult to manage.⁷ Staff wanted to deliver person-centred care, but struggled to do this when feeling overwhelmed, unsupported by management, and unsafe or fearful. They faced tensions in deciding how far to go along with a resident’s disorientation or how to separate a person from their behaviour without undermining personhood. In his work on personhood and dementia, Kitwood highlighted the relational dimension of personhood as connected to both ‘cared for’ and ‘carer’ ⁸. Generally however, this has been related to how those caring for people with dementia can enhance or diminish personhood through their responses and ultimately this may result in staff being blamed or seen as the cause of problems by not being person-centred or doing a good enough job.”

Perhaps this has to do with the mixing up of this descriptive study with a planned intervention study further down the line...Nevertheless, clinical and practical implications is important but does not need to fill the main space for the discussion section – also since the article has a section for this.

As described above, we have now taken out the explicit focus on a planned intervention and reduced the overall focus on clinical implications.

If the authors concentrate on explaining the main findings in the discussion section and not on the planned intervention, the background literature references from intervention studies should be removed. However, a lot of relevant studies are cited in the background section (and only a few studies are referred to in this section) as well as theory on person-centred care (Kitwood) which can be further used in the discussion section to deepen the discussion.

Having changed the focus of the discussion away from the planned intervention we also removed from the introduction:

“In a systematic review of RCTs testing non-pharmacological interventions for agitation, Livingston et al (2014) found that group based activities, music therapy by protocol and sensory interventions, decrease agitation whilst the intervention is being delivered, and supervised interventions training staff in communication skills and person-centred care, reduce agitation for up to six months post-intervention.”

We have left some reference to non-pharmacological interventions for agitation as we believe our study has broader relevance to understandings of why these studies have not demonstrated sustained effects in care homes.

Also, the abstract and summary must be slightly changed.

We have amended the conclusions subheading of the abstract to say:

“Responding to agitation expressed by residents was not a linear process and staff faced tensions and dilemmas in deciding how to respond, especially when initial strategies were unsuccessful or when attempts to respond to residents’ needs were inhibited by structural and procedural constraints in the care home.”

We also deleted:

“Interventions should build upon the approaches and strategies already being used by care home staff in routine care and address constraints”

In the article summary we have changed the point:

- Understanding how staff understand and manage agitation and what makes this harder or easier has informed the development of a psychosocial intervention that reflects and addresses the complexity of delivering interventions in care home settings.

To state:

- Understanding how staff understand and manage agitation and what makes this harder or easier can inform the development of a psychosocial intervention that reflects and addresses the complexity of delivering interventions in care home settings.

In the discussion section a new aim of the article was added (not outlined in the introduction section), to offer an explanatory model – despite that the article did not offer a theory or model in the method section (probably because the study is based on an inductive model). However, a descriptive inductive qualitative study can use theory as well as background literature to further explain or understand its findings, see comments above.

We have removed this secondary aim from the discussion.

Minor comments

It is not stated how the researchers collected informed consent, how anonymity was secured or other research ethical guidelines were followed despite a stated ethical approval.

In order to accommodate additional detail on the informed consent and anonymity were ensured, in line with the ethical approval, we included a figure outlining these procedures in the previous version (See Figure 1 below).

Figure 1: Recruitment and data collection procedures

We hope that you feel that the revisions made have improved the overall quality and clarity of the article in what we feel is both an academically and clinically important topic.

Dr Penny Rapaport (on behalf of all the authors)

References

1. Moore GF, Audrey S, Barker M, et al. Process evaluation of complex interventions: Medical Research Council guidance. *BMJ* 2015;350:h1258. doi: 10.1136/bmj.h1258
2. Geertz C. *Thick Description: Towards an Interpretative Theory of Culture. The Interpretation of Cultures: Selected Essays.* New York: Basic Books 1973:3-30.
3. Livingston G, Barber J, Marston L, et al. Prevalence of and associations with agitation in residents with dementia living in care homes: MARQUE cross-sectional study. *BJPsych Open* 2017;3(4):171-78. doi: 10.1192/bjpo.bp.117.005181
4. Algase DL, Beck C, Kolanowski A, et al. Need-driven dementia-compromised behavior: An alternative view of disruptive behavior. *American Journal of Alzheimer's Disease* 1996;11(6):10-19. doi: 10.1177/153331759601100603
5. Kitwood TM. *Dementia reconsidered: The person comes first:* Open university press 1997.
6. Cohen-Mansfield J, Dakheel-Ali M, Marx MS, et al. Which unmet needs contribute to behavior problems in persons with advanced dementia? *Psychiatry research* 2015;228(1):59-64.
7. Brodaty H, Draper B, Low LF. Nursing home staff attitudes towards residents with dementia: strain and satisfaction with work. *J Adv Nurs* 2003;44(6):583-90. doi: 10.1046/j.0309-2402.2003.02848.x
8. Kitwood T, Bredin K. Towards a theory of dementia care: personhood and well-being. *Ageing Soc* 1992;12(03):269-87. doi: 10.1017/S0144686X0000502X